# Swapping Autoencoder for Deep Image Manipulation

**Taesung Park**[12]     **Jun-Yan Zhu**[2]     **Oliver Wang**[2]     **Jingwan Lu**[2]

**Eli Shechtman**[2]     **Alexei A. Efros**[12]     **Richard Zhang**[2]

[1]UC Berkeley     [2]Adobe Research

## Abstract

Deep generative models have become increasingly effective at producing realistic images from randomly sampled seeds, but using such models for *controllable manipulation of existing images* remains challenging. We propose the Swapping Autoencoder, a deep model designed specifically for image manipulation, rather than random sampling. The key idea is to encode an image into two independent components and enforce that any swapped combination maps to a realistic image. In particular, we encourage the components to represent structure and texture, by enforcing one component to encode co-occurrent patch statistics across different parts of the image. As our method is trained with an encoder, finding the latent codes for a new input image becomes trivial, rather than cumbersome. As a result, our method enables us to manipulate real input images in various ways, including texture swapping, local and global editing, and latent code vector arithmetic. Experiments on multiple datasets show that our model produces better results and is substantially more efficient compared to recent generative models.

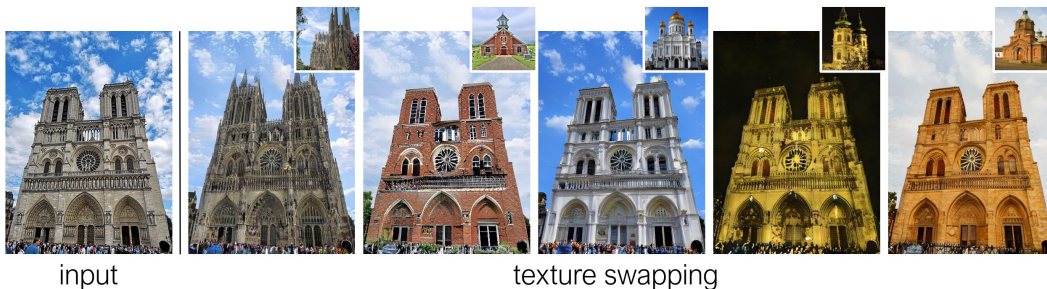

input                                          texture swapping

Figure 1: Our Swapping Autoencoder learns to disentangle texture from structure for image editing tasks. One such task is texture swapping, shown here. Please see our project webpage for a demo video of our editing method.

## 1   Introduction

Traditional photo-editing tools, such as Photoshop, operate solely within the confines of the input image, i.e. they can only "recycle" the pixels that are already there. The promise of using machine learning for image manipulation has been to incorporate the *generic visual knowledge* drawn from external visual datasets into the editing process. The aim is to enable new class of editing operations, such as inpainting large image regions [60, 81, 55], synthesizing photorealistic images from layouts [33, 73, 59], replacing objects [88, 28], or changing the time photo is taken [41, 2].

However, learning-driven image manipulation brings in its own challenges. For image editing, there is a fundamental conflict: what information should be gleaned from the dataset versus information that must be retained from the input image? If the output image relies too much on the dataset, it will retain no resemblance to the input, so can hardly be called "editing", whereas relying too much on the input lessens the value of the dataset. This conflict can be viewed as a disentanglement

problem. Starting from image pixels, one needs to factor out the visual information which is specific to a given image from information that is applicable across different images of the dataset. Indeed, many existing works on learning-based image manipulation, though not always explicitly framed as learning disentanglement, end up doing so, using paired supervision [70, 33, 73, 59], domain supervision [88, 30, 56, 2], or inductive bias of the model architecture [1, 21].

In our work, we aim to discover a disentanglement suitable for image editing in an *unsupervised setting*. We argue that it is natural to explicitly factor out the visual patterns within the image that must change consistently with respect to each other. We operationalize this by learning an autoencoder with two modular latent codes, one to capture the *within-image visual patterns*, and another to capture the rest of the information. We enforce that any arbitrary combination of these codes map to a realistic image. To disentangle these two factors, we ensure that all image patches with the same within-image code appear coherent with each other. Interestingly, this coincides with the classic definition of visual texture in a line of works started by Julesz [38, 40, 39, 64, 24, 17, 54]. The second code captures the remaining information, coinciding with structure. As such, we refer to the two codes as *texture* and *structure* codes.

A natural question to ask is: why not simply use unconditional GANs [19] that have been shown to disentangle style and content in unsupervised settings [43, 44, 21]? The short answer is that these methods do not work well for editing *existing* images. Unconditional GANs learn a mapping from an easy-to-sample (typically Gaussian) distribution. Some methods [4, 1, 44] have been suggested to *retrofit* pre-trained unconditional GAN models to find the latent vector that reproduces the input image, but we show that these methods are inaccurate and magnitudes slower than our method. The conditional GAN models [33, 88, 30, 59] address this problem by starting with input images, but they require the task to be defined *a priori*. In contrast, our model learns an embedding space that is useful for image manipulation in several downstream tasks, including synthesizing new image hybrids (see Figure 1), smooth manipulation of attributes or domain transfer by traversing latent directions (Figure 7), and local manipulation of the scene structure (Figure 8).

To show the effectiveness of our method, we evaluate it on multiple datasets, such as LSUN churches and bedrooms [80], FlickrFaces-HQ [43], and newly collected datasets of mountains and waterfalls, using both automatic metrics and human perceptual judgments. We also present an interactive UI (please see our video in the project webpage) that showcases the advantages of our method.

## 2 Related Work

**Conditional generative models**, such as image-to-image translation [33, 88], learn to directly synthesize an output image given a user input. Many applications have been successfully built with this framework, including image inpainting [60, 32, 77, 55], photo colorization [83, 50, 85, 23], texture and geometry synthesis [86, 20, 75], sketch2photo [66], semantic image synthesis and editing [73, 63, 10, 59]. Recent methods extent it to multi-domain and multi-modal setting [30, 89, 56, 82, 12]. However, it is challenging to apply such methods to on-the-fly image manipulation, because for each new application and new user input, a new model needs to be trained. We present a framework for both image synthesis and manipulation, in which the task can be defined by one or a small number of examples at run-time. While recent works [67, 68] propose to learn a single-image GANs for image editing, our model can be quickly applied to a test image without extensive computation of single-image training.

**Deep image editing via latent space exploration** modifies the latent vector of a pre-trained, unconditional generative model (e.g., a GAN [19]) according to the desired user edits. For example, iGAN [87] obtains the latent code using an encoder-based initialization followed by Quasi-Newton optimization, and updates the code according to new user constraints. Similar ideas have been explored in other tasks like image inpainting, face editing, and deblurring [8, 61, 78, 3]. More recently, instead of using the input latent space, GANPaint [4] adapts layers of a pre-trained GAN for each input image and updates layers according to a user's semantic control [5]. Image2StyleGAN [1] and StyleGAN2 [44] reconstruct the image using an extended embedding space and noise vectors. Our work differs in that we allow the code space to be learned rather than sampled from a fixed distribution, thus making it much more flexible. In addition, we train an encoder together with the generator, which allows for significantly faster reconstruction.

**Disentanglement of content and style generative models.** Deep generative models learn to model the data distribution of natural images [65, 19, 47, 13, 11, 76], many of which aim to represent content and style as independently controllable factors [43, 44, 45, 15, 74]. Of special relevance to our work are models that use code swapping during training [58, 29, 36, 69, 15, 43]. Our work differs from them

in three aspects. First, while most require human supervision, such as class labels [58], pairwise image similarity [36], images pairs with same appearances [15], or object locations [69], our method is fully unsupervised. Second, our decomposable structure and texture codes allow each factor be extracted from the input images to control different aspects of the image, and produce higher-quality results when mixed. Note that for our application, image quality and flexible control are critically important, as we focus on image manipulation rather than unsupervised feature learning. Recent image-to-image translation methods also use code swapping but require ground truth domain labels [49, 51, 53]. In concurrent work, Anokhin et al. [2] and ALAE [62] propose models very close to our code swapping scheme for image editing purposes.

**Style transfer.** Modeling style and content is a classic computer vision and graphics problem [70, 25]. Several recent works revisited the topic using modern neural networks [18, 37, 71, 9], by measuring content using perceptual distance [18, 14], and style as global texture statistics, e.g., a Gram matrix. These methods can transfer low-level styles such as brush strokes, but often fail to capture larger scale semantic structures. Photorealistic style transfer methods further constrain the result to be represented by local affine color transforms from the input image [57, 52, 79], but such methods only allow local color changes. In contrast, our learned decomposition can transfer semantically meaningful structure, such as the architectural details of a church, as well as perform other image editing operations.

## 3 Method

What is the desired representation for image editing? We argue that such representation should be able to reconstruct the input image easily and precisely. Each code in the representation can be independently modified such that the resulting image both looks realistic and reflects the unmodified codes. The representation should also support both global and local image editing.

To achieve the above goals, we train a swapping autoencoder (shown in Figure 2) consisting of an encoder $E$ and a generator $G$, with the core objectives of 1) accurately reconstructing an image, 2) learning independent components that can be mixed to create a new hybrid image, and 3) disentangling texture from structure by using a patch discriminator that learns co-occurrence statistics of image patches.

### 3.1 Accurate and realistic reconstruction

In a classic autoencoder [27], the encoder $E$ and generator $G$ form a mapping between image $\mathbf{x} \sim \mathbf{X} \subset \mathbb{R}^{H \times W \times 3}$ and latent code $\mathbf{z} \sim \mathbf{Z}$. As seen in the top branch of Figure 2, our autoencoder also follows this framework, using an image reconstruction loss:

$$\mathcal{L}_{\text{rec}}(E, G) = \mathbb{E}_{\mathbf{x} \sim \mathbf{X}}[\|\mathbf{x} - G(E(\mathbf{x}))\|_1]. \tag{1}$$

In addition, we wish for the image to be realistic, enforced by a discriminator $D$. The non-saturating adversarial loss [19] for the generator $G$ and encoder $E$ is calculated as:

$$\mathcal{L}_{\text{GAN,rec}}(E, G, D) = \mathbb{E}_{\mathbf{x} \sim \mathbf{X}}[-\log(D(G(E(\mathbf{x}))))]. \tag{2}$$

### 3.2 Decomposable latent codes

We divide the latent space $\mathbf{Z}$ into two components, $\mathbf{z} = (\mathbf{z}_s, \mathbf{z}_t)$, and enforce that swapping components with those from other images still produces realistic images, using the GAN loss [19].

$$\mathcal{L}_{\text{GAN,swap}}(E, G, D) = \mathbb{E}_{\mathbf{x}^1, \mathbf{x}^2 \sim \mathbf{X}, \mathbf{x}^1 \neq \mathbf{x}^2}\left[-\log(D(G(\mathbf{z}_s^1, \mathbf{z}_t^2)))\right], \tag{3}$$

where $\mathbf{z}_s^1$, $\mathbf{z}_t^2$ are the first and second components of $E(\mathbf{x}^1)$, $E(\mathbf{x}^2)$, respectively. Furthermore, as shown in Figure 2, we design the shapes of $\mathbf{z}_s$ and $\mathbf{z}_t$ asymmetrically such that $\mathbf{z}_s$ is a tensor with spatial dimensions, while $\mathbf{z}_t$ is a vector. In our model, $\mathbf{z}_s$ and $\mathbf{z}_t$ are intended to encode structure and texture information, and hence named *structure* and *texture* code, respectively, for convenience. At each training iteration, we randomly sample two images $\mathbf{x}^1$ and $\mathbf{x}^2$, and enforce $\mathcal{L}_{\text{rec}}$ and $\mathcal{L}_{\text{GAN,rec}}$ on $\mathbf{x}^1$, and $\mathcal{L}_{\text{GAN,swap}}$ on the hybrid image of $\mathbf{x}^1$ and $\mathbf{x}^2$.

A majority of recent deep generative models [6, 26, 13, 11, 46, 43, 44], such as in GANs [19] and VAEs [47], attempt to make the latent space Gaussian to enable random sampling. In contrast, we do not enforce such constraint on the latent space of our model. Our swapping constraint focuses on making the "distribution" around a *specific* input image and its plausible variations well-modeled.

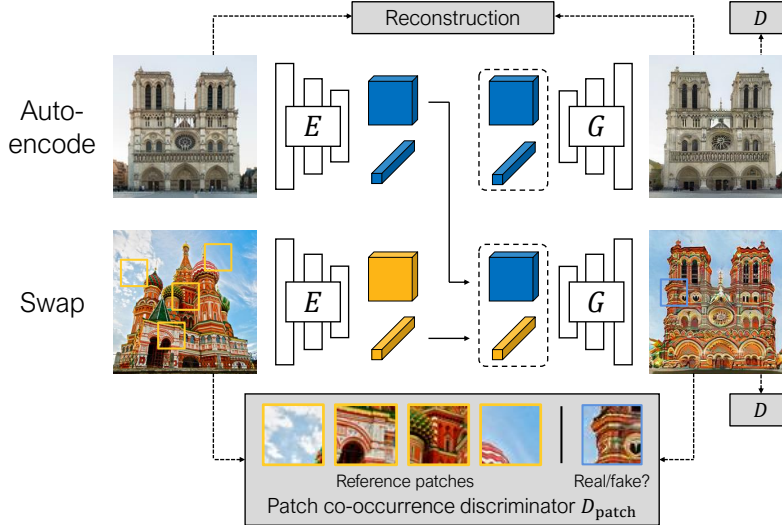

Figure 2: **Swapping Autoencoder** consists of autoencoding (*top*) and swapping (*bottom*) operation. **(Top)** An encoder $E$ embeds an input (Notre-Dame) into two codes. The structure code (□) is a tensor with spatial dimensions; the texture code (▧) is a 2048-dimensional vector. Decoding with generator $G$ should produce a realistic image (enforced by discriminator $D$) matching the input (reconstruction loss). **(Bottom)** Decoding with the texture code from a second image (Saint Basil's Cathedral) should look realistic (via $D$) and match the texture of the image, by training with a patch co-occurrence discriminator $D_{\text{patch}}$ that enforces the output and reference patches look indistinguishable.

Under ideal convergence, the training of the Swapping Autoencoder encourages several desirable properties of the learned embedding space $\mathbf{Z}$. First, the encoding function $E$ is optimized toward injection, due to the reconstruction loss, in that different images are mapped to different latent codes. Also, our design choices encourage that different codes produce different outputs via $G$: the texture code must capture the texture distribution, while the structure code must capture location-specific information of the input images (see Appendix **??** for more details). Lastly, the joint distribution of the two codes of the swap-generated images is factored by construction, since the structure codes are combined with random texture codes.

## 3.3 Co-occurrent patch statistics

While the constraints above are sufficient for our swapping autoencoder to learn a factored representation, the resulting representation will not necessarily be intuitive for image editing, with no guarantee that $\mathbf{z}_s$ and $\mathbf{z}_t$ actually represent structure and texture. To address this, we encourage the texture code $\mathbf{z}_t$ to maintain the same texture in any swap-generated images. We introduce a patch co-occurrence discriminator $D_{\text{patch}}$, as shown in the bottom of Figure 2. The generator aims to generate a hybrid image $G(\mathbf{z}_s^1, \mathbf{z}_t^2)$, such that any patch from the hybrid cannot be distinguished from a group of patches from input $\mathbf{x}^2$.

$$\mathcal{L}_{\text{CooccurGAN}}(E, G, D_{\text{patch}}) = \mathbb{E}_{\mathbf{x}^1, \mathbf{x}^2 \sim \mathbf{X}}\left[-\log\left(D_{\text{patch}}\left(\texttt{crop}(G(\mathbf{z}_s^1, \mathbf{z}_t^2)), \texttt{crops}(\mathbf{x}^2)\right)\right)\right], \quad (4)$$

where `crop` selects a random patch of size $1/8$ to $1/4$ of the full image dimension on each side (and `crops` is a collection of multiple patches). Our formulation is inspired by Julesz's theory of texture perception [38, 40] (long used in texture synthesis [64, 17]), which hypothesizes that images with similar marginal and joint feature statistics appear perceptually similar. Our co-occurence discriminator serves to enforce that the joint statistics of a learned representation be consistently transferred. Similar ideas for modeling co-occurences have been used for propagating a single texture in a supervised setting [75], self-supervised representation learning [34], and identifying image composites [31].

## 3.4 Overall training and architecture

Our final objective function for the encoder and generator is $\mathcal{L}_{\text{total}} = \mathcal{L}_{\text{rec}} + 0.5\mathcal{L}_{\text{GAN,rec}} + 0.5\mathcal{L}_{\text{GAN,swap}} + \mathcal{L}_{\text{CooccurGAN}}$. The discriminator objective and design follows StyleGAN2 [44]. The co-occurrence patch discriminator first extracts features for each patch, and then concatenates them to pass to the final

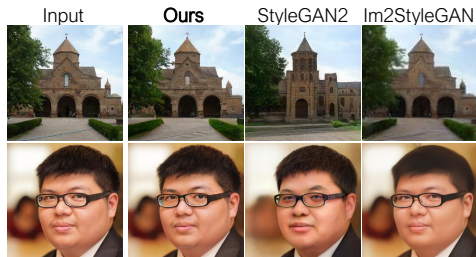

| Method | Runtime (sec) (↓) | LPIPS Reconstruction (↓) | | | |
|---|---|---|---|---|---|
| | | Church | FFHQ | Waterfall | Average |
| Ours | **0.101** | 0.227 | **0.074** | **0.238** | **0.180** |
| Im2StyleGAN | 495 | **0.186** | 0.174 | 0.281 | 0.214 |
| StyleGAN2 | 96 | 0.377 | 0.215 | 0.384 | 0.325 |

Figure 3: **Embedding examples and reconstruction quality**. We project images into embedding spaces for our method and baseline GAN models, Im2StyleGAN [1, 43] and StyleGAN2 [44]. Our reconstructions better preserve the detailed outline (e.g., doorway, eye gaze) than StyleGAN2, and appear crisper than Im2StyleGAN. This is verified on average with the LPIPS metric [84]. Our method also reconstructs images much faster than recent generative models that use iterative optimization. See Appendix **??** for more visual examples.

classification layer. The encoder consists of 4 downsampling ResNet [22] blocks to produce the tensor $\mathbf{z}_s$, and a dense layer after average pooling to produce the vector $\mathbf{z}_t$. As a consequence, the structure code $\mathbf{z}_s$, is limited by its receptive field at each location, providing an inductive bias for capturing local information. On the other hand, the texture code $\mathbf{z}_t$, deprived of spatial information by the average pooling, can only process aggregated feature distributions, forming a bias for controlling global style. The generator is based on StyleGAN2, with weights modulated by the texture code. Please see Appendix **??** for a detailed specification of the architecture, as well as details of the discriminator loss function.

## 4 Experiments

The proposed method can be used to efficiently embed a given image into a factored latent space, and to generate hybrid images by swapping latent codes. We show that the disentanglement of latent codes into the classic concepts of "style" and "content" is competitive even with style transfer methods that address this specific task [48, 79], while producing more photorealistic results. Furthermore, we observe that even without an explicit objective to encourage it, vector arithmetic in the learned embedding space **Z** leads to consistent and plausible image manipulations [7, 43, 35]. This opens up a powerful set of operations, such as attribute editing, image translation, and interactive image editing, which we explore.

We first describe our experimental setup. We then evaluate our method on: (1) quickly and accurately embedding a test image, (2) producing realistic hybrid images with a factored latent code that corresponds to the concepts of texture and structure, and (3) editability and usefulness of the latent space. We evaluate each aspect separately, with appropriate comparisons to existing methods.

### 4.1 Experimental setup

**Datasets.** For existing datasets, our model is trained on LSUN Churches, Bedrooms [80], Animal Faces HQ (AFHQ) [12], Flickr Faces HQ (FFHQ) [43], all at resolution of 256px except FFHQ at 1024px. In addition, we introduce new datasets, which are Portrait2FFHQ, a combined dataset of 17k portrait paintings from `wikiart.org` and FFHQ at 256px, Flickr Mountain, 0.5M mountain images from `flickr.com`, and Waterfall, of 90k 256px waterfall images. Flickr Mountain is trained at 512px resolution, but the model can handle larger image sizes (e.g., 1920×1080) due to the fully convolutional architecture.

**Baselines.** To use a GAN model for downstream image editing, one must embed the image into its latent space [87]. We compare our approach to two recent solutions. Im2StyleGAN [1] present a method for embedding into StyleGAN [43], using iterative optimization into the "$W^+$-space" of the model. The StyleGAN2 model [44] also includes an optimization-based method to embed into its latent space and noise vectors. One application of this embedding is producing hybrids. StyleGAN and StyleGAN2 present an emergent hierarchical parameter space that allows hybrids to be produced by mixing parameters of two images. We additionally compare to image stylization methods, which aim to mix the "style" of one image with the "content" from another. STROTSS [48] is an optimization-based framework, in the spirit of the classic method of Gatys et al. [18]. We also compare to WCT$^2$ [79], a recent state-of-the-art *photorealistic* style transfer method based on a feedforward network.

### 4.2 Image embedding

The first step of manipulating an image with a generative model is projecting it into its latent spade. If the input image cannot be projected with high fidelity, the embedded vector cannot be used for editing, as the user would be editing a different image. Figure 3 illustrates both example reconstructions

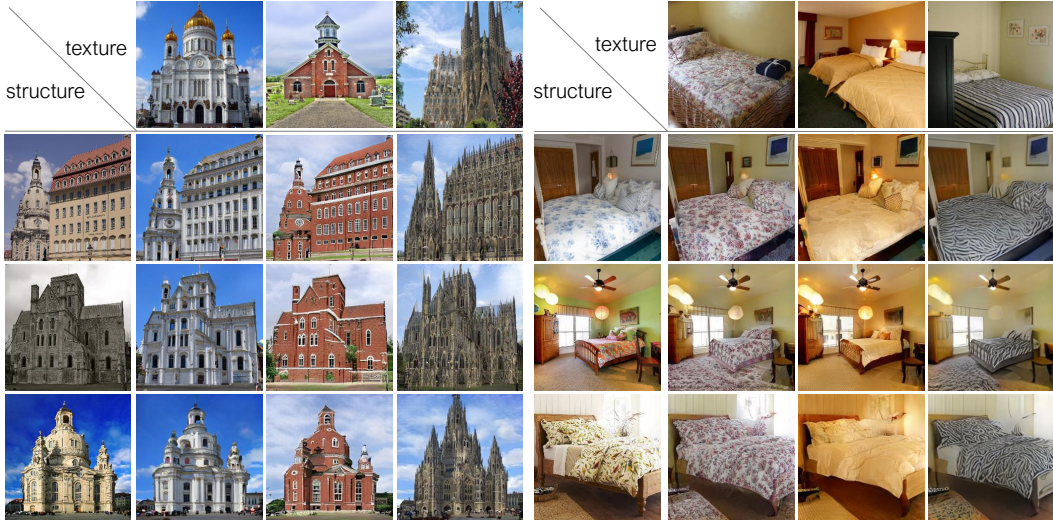

Figure 4: **Image swapping**. Each row shows the result of combining the structure code of the leftmost image with the texture code of the top image (trained on LSUN Church and Bedroom). Our model generates realistic images that preserve texture (e.g., material of the building, or the bedsheet pattern) and structure (outline of objects).

| Method | Runtime (sec) (↓) | Human Perceptual Study (AMT Fooling Rate) (↑) | | | |
| --- | --- | --- | --- | --- | --- |
| | | Church | FFHQ | Waterfall | **Average** |
| Swap Autoencoder (Ours) | **0.113** | **31.3±2.4** | **19.4±2.0** | **41.8±2.2** | **31.0±1.4** |
| Im2StyleGAN [1, 43] | 990 | 8.5±2.1 | 3.9±1.1 | 12.8±2.4 | 8.4±1.2 |
| StyleGAN2 [44] | 192 | 24.3±2.2 | 13.8±1.8 | 35.3±2.4 | 24.4±1.4 |
| STROTSS [48] | 166 | 13.7±2.2 | 3.5±1.1 | 23.0±2.1 | 13.5±1.2 |
| WCT$^2$ [79] | 1.35 | *27.9±2.3* | 22.3±2.0 | 35.8±2.4 | *28.6±1.3* |

Table 1: **Realism of swap-generated images** We study how realistic our swap-generated swapped appear, compared to state-of-the-art generative modeling approaches (Im2StyleGAN and StyleGAN2) and stylization methods (STROTSS and WCT$^2$). We run a perceptual study, where each method/dataset is evaluated with 1000 human judgments. We **bold** the best result per column and ***bold+italicize*** methods that are within the statistical significance of the top method. Our method achieves the highest score across all datasets. Note that WCT$^2$ is a method tailored especially for photorealistic style transfer and is within the statistical significance of our method in the perceptual study. Runtime is reported for 1024×1024 resolution.

and quantitative measurement of reconstruction quality, using LPIPS [84] between the original and embedded images. Note that our method accurately preserves the doorway pattern (top) and facial features (bottom) without blurriness. Averaged across datasets and on 5 of the 6 comparisons to the baselines, our method achieves *better reconstruction quality* than the baselines. An exception is on the Church dataset, where Im2StyleGAN obtains a better reconstruction score. Importantly, as our method is designed with test-time embedding in mind, it only requires a single feedforward pass, at least 1000× faster than the baselines that require hundreds to thousands of optimization steps. Next, we investigate how *useful* the embedding is by exploring manipulations with the resulting code.

### 4.3 Swapping to produce image hybrids

In Figure 4, we show example hybrid images with our method, produced by combining structure and texture codes from different images. Note that the textures of the top row of images are consistently transferred; the sky, facade, and window patterns are mapped to the appropriate regions on the structure images on the churches, and similarly for the bedsheets.

**Realism of image hybrids.** In Table 1, we show results of comparison to existing methods. As well as generative modeling methods [1, 44, 43]. For image hybrids, we additionally compare with SOTA style transfer methods [48, 79], although they are not directly applicable for controllable editing by embedding images (Section 4.5). We run a human perceptual study, following the test setup used in [83, 33, 67]. A real and generated image are shown sequentially for one second each to Amazon Mechanical Turkers (AMT), who choose which they believe to be fake. We measure how often they fail to identify the fake. An algorithm generating perfectly plausible images would achieve a fooling

| Structure | Texture | | StyleGAN2 | Im2StyleGAN | STROTSS | WCT² | **Ours** |
|---|---|---|---|---|---|---|---|

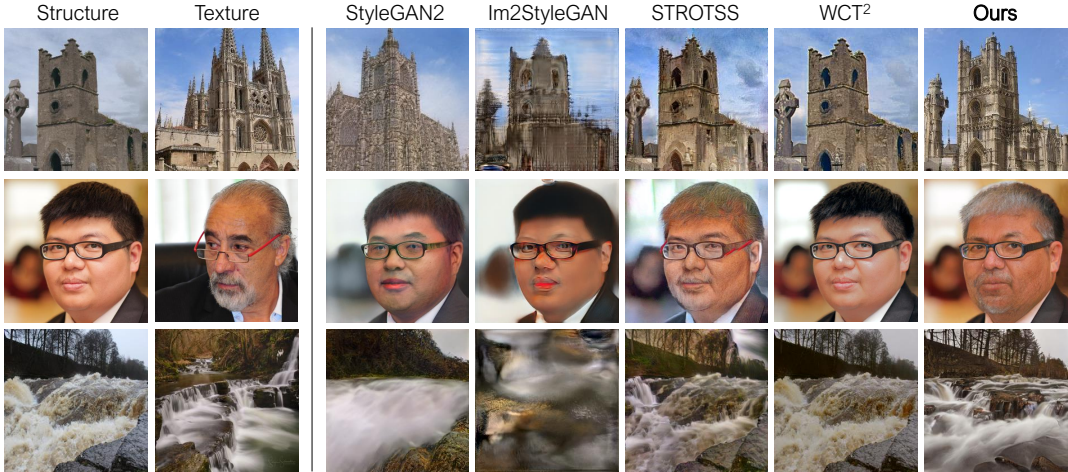

Figure 5: **Comparison of image hybrids.** Our approach generates realistic results that combine scene structure with elements of global texture, such as the shape of the towers (church), the hair color (portrait), and the long exposure (waterfall). Please see Appendix **??** for more comparisons.

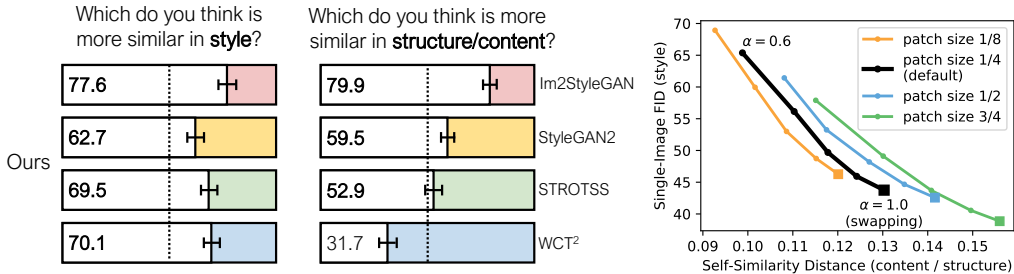

Figure 6: **Style and content**. **(Left)** Results of our perceptual study where we asked users on AMT to choose which image better reflects the "style" or "content" of a provided reference image, given two results (ours and a baseline). Our model is rated best for capturing style, and second-best for preserving content, behind WCT² [79], a photorealistic style transfer method. Most importantly, our method was rated strictly better in both style and content matching than both image synthesis models Im2StyleGAN [1, 43] and StyleGAN2 [44]. **(Right)** Using the self-similarity distance [48] and SIFID [67], we study variations of the co-occurrence discriminator's patch size in training with respect to the image size. As patch size increases, our model tends to make more changes in swapping (closer to the target style and further from input structure). In addition, we gradually interpolate the texture code, with interpolation ratio $\alpha$, away from a full swapping $\alpha = 1.0$, and observe that the transition is smooth.

rate of $50\%$. We gather 15,000 judgments, 1000 for each algorithm and dataset. Our method achieves more realistic results across all datasets. The nearest competitor is the WCT² [79] method, which is designed for photorealistic style transfer. Averaged across the three datasets, our method achieves the highest fooling rate ($31.0\pm1.4\%$), with WCT² closely following within the statistical significance ($28.6\pm1.3\%$). We show qualitative examples in Figure 5.

**Style and content.** Next, we study how well the concepts of *content* and *style* are reflected in the structure and texture codes, respectively. We employ a Two-alternative Forced Choice (2AFC) user study to quantify the quality of image hybrids in content and style space. We show participants our result and a baseline result, with the style or content reference in between. We then ask a user which image is more similar in style, or content respectively. Such 2AFC tests were used to train the LPIPS perceptual metric [84], as well as to evaluate style transfer methods in [48]. As no true automatic perceptual function exists, human perceptual judgments remain the "gold standard" for evaluating image synthesis results [83, 33, 10, 67]. Figure 6 visualizes the result of 3,750 user judgments over four baselines and three datasets, which reveal that our method outperforms all baseline methods with statistical significance in style preservation. For content preservation, our method is only behind WCT², which is a photorealistic stylization method that makes only minor color modifications to the input. Most importantly, our method achieves the best performance with statistical significance in both style and content among models that can embed images, which is required for other forms of image editing.

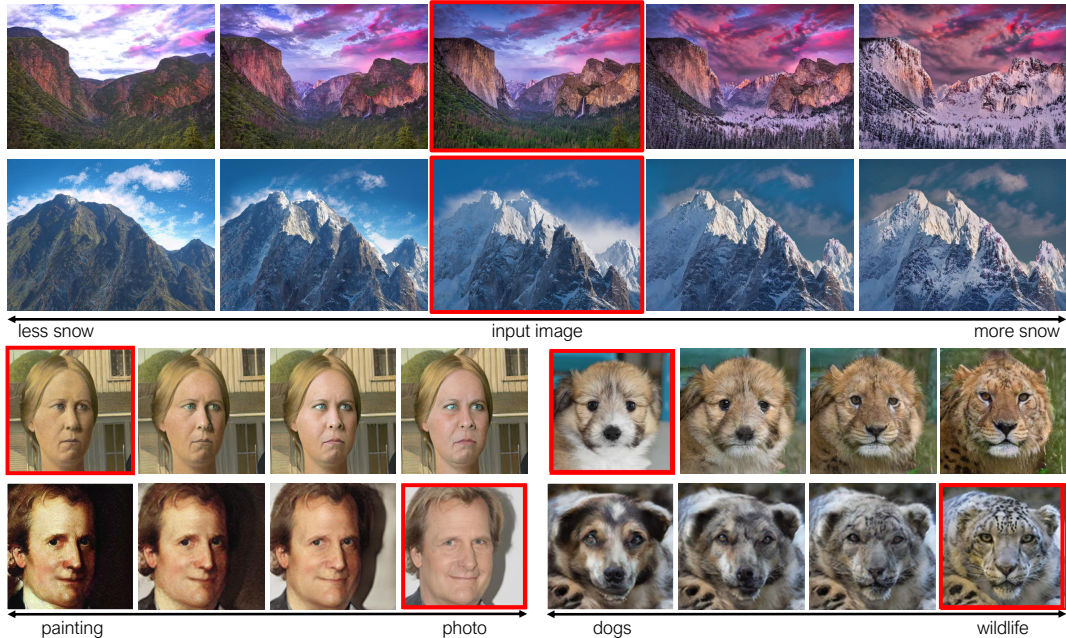

Figure 7: **Continuous interpolation**. **(top)** A manipulation vector for *snow* is discovered by taking mean difference between 10 user-collected photos of snowy and summer mountain. The vector is simply added to the texture code of the input image (red) with some gain. **(bottom)** Multi-domain, continuous transformation is achieved by applying the average vector difference between the texture codes of two domains, based on annotations from the training sets. We train on Portrait2FFHQ and AFHQ [12] datasets. See Appendix **??** for more results.

## 4.4 Analysis of our method

Next we analyze the behavior of our model using automated metrics. Self-similarity Distance [48] measures structural similarity in deep feature space based on the self-similarity map of ImageNet-pretrained network features. Single-Image FID [67] measures style similarity by computing the Fréchet Inception Distance (FID) between two feature distributions, each generated from a single image. SIFID is similar to Gram distance, a popular metric in stylization methods [18, 17], but differs by comparing the mean of the feature distribution as well as the covariance.

Specifically, we vary the size of cropped patches for the co-occurrence patch discriminator in training. In Figure 6 *(right)*, the max size of random cropping is varied from $1/8$ to $3/4$ of the image side length, including the default setting of $1/4$. We observe that as the co-occurrence discriminator sees larger patches, it enforces stronger constraint, thereby introducing more visual change in both style and content. Moreover, instead of full swapping, we gradually interpolate one texture code to the other. We observe that the SIFID and self-similarity distance both change gradually, in all patch settings. Such gradual visual change can be clearly observed in Figure 7, and the metrics confirm this.

## 4.5 Image editing via latent space operations

Even though no explicit constraint was enforced on the latent space, we find that modifications to the latent vectors cause smooth and predictable transformations to the resulting images. This makes such a space amenable to downstream editing in multiple ways. First, we find that our representation allows for controllable image manipulations by **vector arithmetic** in the latent space. Figure 7 shows that adding the same vector smoothly transforms different images into a similar style, such as gradually adding more snow *(top)*. Such vectors can be conveniently derived by taking the mean difference between the embeddings of two groups of images.

In a similar mechanism, the learned embedding space can also be used for **image-to-image translation** tasks (Figure 7), such as transforming paintings to photos. Image translation is achieved by applying the domain translation vector, computed as the mean difference between the two domains. Compared to most existing image translation methods, our method does not require that all images are labeled, and also allows for multi-domain, fine-grained control simply by modifying the vector magnitude and

members of the domain at test time. Finally, the design of the structure code $\mathbf{z}_s$ is directly amenable **local editing** operations, due to its spatial nature; we show additional results in Appendix **??**.

## 4.6 Interactive user interface for image editing

Based on the proposed latent space exploration methods, we built a sample user interface for creative user control over photographs. Figure 8 shows three editing modes that our model supports. Please see a demo video on our webpage. We demonstrate three operations: (1) **global style editing**: the texture code can be transformed by adding predefined manipulation vectors that are computed from PCA on the train set. Like GANSpace [21], the user is provided with knobs to adjust the gain for each manipulation vector. (2) **region editing**: the structure code can also be manipulated the same way of using PCA components, by treating each location as individual, controllable vectors. In addition, masks can be automatically provided to the user based on the self-similarity map at the location of interest to control the extent of structural manipulation. (3) **cloning**: the structure code can be directly edited using a brush that replaces the code from another part of the image, like the Clone Stamp tool of Photoshop.

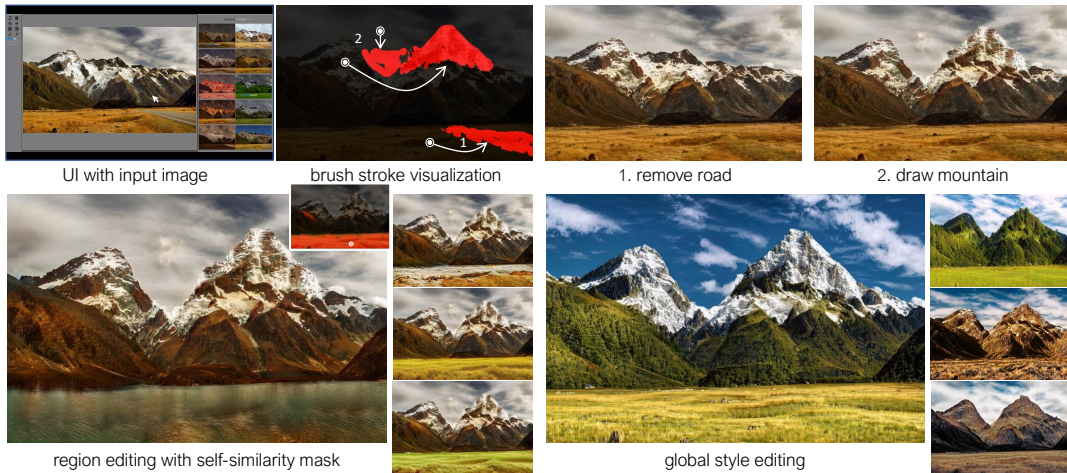

| UI with input image | brush stroke visualization | 1. remove road | 2. draw mountain |

region editing with self-similarity mask          global style editing

Figure 8: **Example Interactive UI. (top, cloning)** using an interactive UI, part of the image is "redrawn" by the user with a brush tool that extracts structure code from user-specified location. **(left, region editing)** the bottom region is transformed to lake, snow, or different vegetation by adding a manipulation vector to the structure codes of the masked region, which is auto-generated from the self-similarity map at the specified location. **(right, global style editing)** the overall texture and style can be changed using vector arithmetic with principal directions of PCA, controlled by the sliders on the right pane of the UI. (best viewed zoomed in)

## 5 Discussion

The main question we would like to address, is whether unconditional random image generation is required for high-quality image editing tasks. For such approaches, projection becomes a challenging operation, and intuitive disentanglement still remains a challenging question. We show that our method based on an auto-encoder model has a number of advantages over prior work, in that it can accurately embed high-resolution images in real-time, into an embedding space that disentangles texture from structure, and generates realistic output images with both swapping and vector arithmetic. We performed extensive qualitative and quantitative evaluations of our method on multiple datasets. Still, structured texture transfer remains challenging, such as the striped bedsheet of Figure 4. Furthermore, extensive analysis on the nature of disentanglement, ideally using reliable, automatic metrics will be beneficial as future work.

**Acknowledgments.** We thank Nicholas Kolkin for the helpful discussion on the automated content and style evaluation, Jeongo Seo and Yoseob Kim for advice on the user interface, and William T. Peebles, Tongzhou Wang, and Yu Sun for the discussion on disentanglement.

## Broader Impact

From the sculptor's chisel to the painter's brush, tools for creative expression are an important part of human culture. The advent of digital photography and professional editing tools, such as Adobe Photoshop, has allowed artists to push creative boundaries. However, the existing tools are typically too complicated to be useful by the general public. Our work is one of the new generation of visual content creation methods that aim to democratize the creative process. The goal is to provide intuitive controls (see Section 4.6) for making a wider range of realistic visual effects available to non-experts.

While the goal of this work is to support artistic and creative applications, the potential misuse of such technology for purposes of deception – posing generated images as real photographs – is quite concerning. To partially mitigate this concern, we can use the advances in the field of image forensics [16], as a way of verifying the authenticity of a given image. In particular, Wang et al. [72] recently showed that a classifier trained to classify between real photographs and synthetic images generated by ProGAN [42], was able to detect fakes produced by other generators, among them, StyleGAN [43] and Style-GAN2 [44]. We take a pretrained model of [72] and report the detection rates on several datasets in Appendix ??. Our swap-generated images can be detected with an average rate greater than 90%, and this indicates that our method shares enough architectural components with previous methods to be detectable. However, these detection methods do not work at 100%, and performance can degrade as the images are degraded in the wild (e.g., compressed, rescanned) or via adversarial attacks. Therefore, the problem of verifying image provenance remains a significant challenge to society that requires multiple layers of solutions, from technical (such as learning-based detection systems or authenticity certification chains), to social, such as efforts to increase public awareness of the problem, to regulatory and legislative.

## Funding Disclosure

Taesung Park is supported by a Samsung Scholarship and an Adobe Research Fellowship, and much of this work was done as an Adobe Research intern. This work was supported in part by an Adobe gift.

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
