[Supplementary Material]

# Supplementary Material:
# Swapping Autoencoder for Deep Image Manipulation

Here, we discuss the implementation details, provide details on the factorization of our representation, and show additional results.

## 1 Results and Comparisons

### 1.1 Additional visual results

In Figure 1, 4, and 7 of the main paper, we have shown our results of swapping the texture and structure codes, and manipulation results of the latent space. Here we show more swapping and editing results.

**Swapping.** Here we show additional results of swapping on FFHQ (Figure 1), Mountains (Figure 4), and LSUN Church and Bedroom (Figure 6) dataset. For test images, the input images for the models trained on FFHQ (Figure 1, 2, and 3) and Mountains (Figure 4 and 5) are separately downloaded from `pixabay.com` using relevant keywords. The results on LSUN (Figure 6) are from the validation sets (18).

**Editing.** The latent space of our method can be used for image editing. For example, in Figure 3 and 5, we show the result of editing the texture code using an interactive UI that performs vector arithmetic using the PCA components. Editing the texture code results in changing global attributes like age, wearing glasses, lighting, and background in the FFHQ dataset (Figure 3), and time of day and grayscale in the Mountains dataset (Figure 5). On the other hand, editing the structure code can manipulate locally isolated attributes such as eye shape, gaze direction (Figure 2), or texture of the grass field (Figure 5). These results are generated by performing vector arithmetic in the latent space of the flattened structure code, masked by the region specified by the user in the UI (region editing of Figure 8 of the main paper). In addition, the pond of Figure 5 is created by overwriting the structure code with the code of a lake from another image. More editing results of using the interactive UI can be found on our project webpage: `https://taesungp.github.io/SwappingAutoencoder`.

**User-guided image translation.** In Figure 8, we show the results of user-guided image translation, trained on Portrait2FFHQ and Animal Faces HQ (2). For each dataset, the results are produced using the model trained on the mix of all domains and hence without any domain labels. By adjusting the gains on the principal components of the texture code with the interactive UI, the user controls the magnitude and style of translation. Interestingly, we found that the first principal axis of the texture code largely corresponds to the domain translation vector in the case of Portrait2FFHQ and AFHQ dataset, with the subsequent vectors controlling more fine-grained styles. Therefore, our model is suitable for the inherent multi-modal nature of image translation. For example, in Figure 8, the input cat and dog images are translated into six different plausible outputs.

### 1.2 Additional comparison to existing methods

In Table 1, we report the FIDs of the swapping results of our model and baselines on LSUN Church, FFHQ, and Waterfall datasets using the validation set. Additional visual comparison results are in Figure 7. Note that using FID to evaluate the results of this task is not sufficient, as it does not capture the relationship to input content and style images. For example, a low FID can be achieved simply by not making large changes to the input content image. Our model achieves the second-best FID, behind the photorealistic style transfer method WCT$^2$ (17). However, the visual results of Figure 7 and human perceptual study of the main paper reveal that our method better captures the details of the reference style. In Table 2, we compare the FIDs of swapping on the training set with *unconditionally* generated StyleGAN and StyleGAN2 outputs. Note that randomly sampled images of StyleGAN and StyleGAN2 are not suitable for image editing, as it ignores the input image. The FID of swap-generated images of our method is placed between the FID of unconditionally generated StyleGAN and StyleGAN2 images.

| Method | Church | FFHQ | Waterfall | Mean |
|---|---|---|---|---|
| Swap Autoencoder (Ours) | 52.34 | 59.83 | 50.90 | 54.36 |
| Im2StyleGAN (1; 8) | 219.50 | 123.13 | 267.25 | 203.29 |
| StyleGAN2 (9) | 57.54 | 81.44 | 57.46 | 65.48 |
| STROTSS (11) | 70.22 | 92.19 | 108.41 | 83.36 |
| WCT$^2$ (17) | 35.65 | 39.02 | 35.88 | 36.85 |

Table 1: **FID of swapping on the validation set**. We compare the FIDs of content-style mixing on the validation sets. Note the utility of FID is limited in our setting, since it does not capture the quality of embedding or disentanglement. Our method achieves second-lowest FID, behind WCT$^2$ (17), a photorealistic style transfer method. Note that the values are not directly comparable to different datasets or to the training splits (Table 2), since the number of samples are different. Please see Figure 7 for visual results.

| Method | Church | FFHQ | Waterfall |
|---|---|---|---|
| Swap Autoencoder (Ours) | 3.91 | 3.48 | 3.04 |
| StyleGAN (8) | 4.21 | 4.40* | 6.09 |
| StyleGAN2 (9) | 3.86* | 2.84* | 2.67 |

Table 2: **FID of swapping on the training set, in the context of unconditional GAN**. We compute the FID of swapped images on the training set, and compare it with FIDs of unconditionally generated images of StyleGAN (8) and StyleGAN2 (9). The result conveys how much realism the swap-generated images convey. Note that randomly sampled images of StyleGAN (8) and StyleGAN2 (9) models are not suitable for image editing. Asterisk(*) denotes FIDs reported in the original papers.

Figure 1: **Swapping results of our FFHQ model**. The input photographs are collected from `pixabay.com`.

input                 bigger eyes           gaze direction        more smile            5 o'clock shadow

Figure 2: **Region editing**. The results are generated by performing vector arithmetic on the structure code. The vectors are discovered by a user with our UI, with each goal in mind.

input                 age                   glasses               lighting              background

Figure 3: **Global editing**. The results are generated using vector arithmetic on the texture code. The vectors are discovered by a user with our UI, with each goal in mind.

Figure 4: **Swapping results of our method trained on Flickr Mountains**. The model is trained and tested at 512px height.

Figure 5: **User editing results of our method trained on Flickr Mountains**. For the input image in red, the top and bottom rows show examples of editing the structure and texture code, respectively. Please refer to Figure **??** on how editing is performed. The image is of 1536×1020 resolution, using a model trained at 512px resolution.

Figure 6: **Swapping results of LSUN** Churches (top) and Bedrooms (bottom) validation set. The model is trained with 256px-by-256px crops and tested at 256px resolution on the shorter side, keeping the aspect ratio.

Figure 7: **Comparison to existing methods**. Random results on LSUN Churches and Flickr Waterfall are shown. In each block, we show both the reconstruction and swapping for ours, Im2StyleGAN (1; 8), and StyleGAN2 (9), as well as the style transfer results of STROTSS (11) and WCT$^2$ (17). Im2StyleGAN has a low reconstruction error but performs poorly on the swapping task. StyleGAN2 generates realistic swappings, but fails to capture the input images faithfully. Both style transfer methods makes small changes to the input structure images.

Figure 8: **User-guided image translation**. Using the interactive UI, the user controls the magnitude and style of the translated image. We show the edit results of turning paintings into photo (top) on the model trained on the Portrait2FFHQ dataset, and translating within the Animal Faces HQ dataset (bottom). The input images are marked in red. For the animal image translation, 6 different outputs are shown for the same input image.

## 1.3 Corruption study of Self-Similarity Distance and SIFID

In Figure 9, we validate our usage of Self-Similarity Matrix Distance (11) and Single-Image FID (SIFID) (14) as automated metrics for measuring distance in structure and style. Following FID (5), we study the change in both metrics under predefined corruptions. We find that the self-similarity distance shows a larger variation for image translation and rotation than blurring or adding white noise. In contrast, SIFID is more sensitive to blurring or white noise than translation or rotation. This confirms that the self-similarity captures structure, and SIFID captures style.

Figure 9: **Validating the Self-Similarity Matrix Distance and Single-Image FID**. We apply different types of corruptions and study the variation in the Self-Similarity Distance (11) and Single-Image FID (14). SIFID shows higher sensitivity to overall style changes, such as Gaussian noise or blurring, than structural changes, such as shift and rotation. On the other hand, Self-Similarity Distance shows higher variation for structural changes. This empirically confirms our usage of the two metrics as measuring distance in structure and style.

## 2  Implementation details

We show our architecture designs, additional training details, and provide information about our datasets.

### 2.1  Architecture

The **encoder** maps the input image to structure and texture codes, as shown in Figure 10 (left). For the structure code, the network consists of 4 downsampling residual blocks (4), followed by two convolution layers. For the texture code, the network branches off and adds 2 convolutional layers, followed by an average pooling (to completely remove spatial dimensions) and a dense layer. The asymmetry of the code shapes is designed to impose an inductive bias and encourage decomposition into orthogonal tensor dimensions. Given an $256 \times 256$ image, the structure code is of dimension $16 \times 16 \times 8$ (large spatial dimension), and texture code is of dimension $1 \times 1 \times 2048$ (large channel dimension).

The texture code is designed to be agnostic to positional information by using reflection padding or no padding ("valid") in the convolutional layers (rather than zero padding) followed by average pooling. On the other hand, each location of the structure code has a strong inductive bias to encode information in its neighborhood, due to its fully convolutional architecture and limited receptive field.

The **generator** maps the codes back to an image, as shown in Figure 10 (right). The network uses the structure code in the main branch, which consists of 4 residual blocks and 4 upsampling residual blocks. The texture code is injected using the weight modulation/demodulation layer from StyleGAN2 (9). We generate the output image by applying a convolutional layer at the end of the residual blocks. This is different from the default setting of StyleGAN2, which uses an output skip, but more similar to the residual net setting of StyleGAN2 discriminator. Lastly, to enable isolated local editing, we avoid normalizations such as instance or batch normalization (15; 6).

The **discriminator** architecture is identical to StyleGAN2, except with no minibatch discrimination, to enable easier fine-tuning at higher resolutions with smaller batch sizes.

The **co-occurrence patch discriminator** architecture is shown in Figure 11 and is designed to determine if a patch in question ("real/fake patch") is from the same image as a set of reference patches. Each patch is first independently encoded with 5 downsampling residual blocks, 1 residual block, and 1 convolutional layer. The representations for the reference patches are averaged together and concatenated with the representation of the real/fake patch. The classification applies 3 dense layers to output the final prediction.

The detailed design choices of the layers in all the networks follow StyleGAN2 (9), including weight demodulation, antialiased bilinear down/upsampling (19), equalized learning rate, noise injection at every layer, adjusting variance of residual blocks by the division of $\sqrt{2}$, and leaky ReLU with slope $0.2$.

### 2.2  Training details

At each iteration, we sample a minibatch of size $N$ and produce $N/2$ reconstructed images and $N/2$ hybrid images. The reconstruction loss is computed using $N/2$ reconstructed images. The loss for the image discriminator is computed on the real, reconstructed, and hybrid images, using the adversarial loss $\mathbb{E}\left[-\log(D(\mathbf{x}))\right] + \mathbb{E}\left[-\log(1-D(\mathbf{x}_{\text{fake}}))\right]$, where $\mathbf{x}$ and $\mathbf{x}_{\text{fake}}$ are real and generated (both reconstructed and hybrid) images, respectively. For the details of the GAN loss, we follow the setting of StyleGAN2 (9), including the non-saturating GAN loss (3) and lazy R1 regularization (13; 9). In particular, R1 regularization is also applied to the co-occurrence patch discriminator. The weight for R1 regularization was 10.0 for the image discriminator (following the setting of (13; 9)) and 1.0 for the co-occurrence discriminator. Lastly, the co-occurrence patch discriminator loss is computed on random crops of the real and swapped images. The size of the crops are randomly chosen between $1/8$ and $1/4$ of the image dimensions for each side, and are then resized to $1/4$ of the original image. For each image (real or fake), 8 crops are made. For the query image (the first argument to $D_{\text{patch}}$), each crop is used to predict co-occurrence, producing $8N$ predictions at each iteration. For the reference image (the second argument to $D_{\text{patch}}$), the feature outputs are averaged before concatenated with the query feature. Both discriminators use the binary cross-entropy GAN loss.

We use ADAM (10) with 0.002 learning rate, $\beta_1 = 0.0$ and $\beta_2 = 0.99$. We use the maximum batch size that fits in memory on 8 16GB Titan V100 GPUs: 64 for images of $256 \times 256$ resolution, 16 for $512 \times 512$ resolution, and 16 for $1024 \times 1024$ resolution (with smaller network capacity). Note that only

Figure 10: **Encoder and generator architecture**. The encoder network first applies 4 downsampling residual blocks (4) to produce an intermediate tensor, which is then passed to two separate branches, producing the structure code and texture code. The structure code is produced by applying 1-by-1 convolutions to the intermediate tensor. The texture code is produced by applying strided convolutions, average pooling, and then a dense layer. Given an $H \times H$ image, the shapes of the two codes are $H/16 \times H/16 \times 8$, and $1 \times 1 \times 2048$, respectively. The case for a 512×512 image is shown. To prevent the texture code from encoding positional information, we apply reflection padding for the residual blocks, and then no padding for the conv blocks. The generator consists of 4 residual blocks and then 4 upsampling residual blocks, followed by 1-by-1 convolution to produce an RGB image. The structure code is given in the beginning of the network, and the texture code is provided at every layer as modulation parameters. We use zero padding for the generator. The detailed architecture follows StyleGAN2 (9), including weight demodulation, bilinear upsampling, equalized learning rate, noise injection at every layer, adjusting variance of residual blocks by the division of $\sqrt{2}$, and leaky ReLU with slope 0.2.

Figure 11: **Co-occurrence patch discriminator architecture**. The co-occurrence patch discriminator consists of the feature extractor, which applies 5 downsampling residual blocks, 1 residual block, and 1 convolutional layer with valid padding to each input patch, and the classifier, which concatenates the flattened features in channel dimension and then applies 3 dense layers to output the final prediction. Since the patches have random sizes, they are upscaled to the same size before passed to the co-occurrence discriminator. All convolutions use kernel size 3×3. Residual blocks use the same design as those of the image discriminator. For the reference patches, more than one patch is used, so the extracted features are averaged over the batch dimension to capture the aggregated distribution of the reference texture.

the FFHQ dataset was trained at 1024×1024 resolution; for the landscape datasets, we take advantage of the fully convolutional architecture and train with cropped images of size 512×512, and test on the full image. The weights on each loss term are simply set to be all 1.0 among the reconstruction, image GAN, and co-occurrence GAN loss.

## 2.3 Datasets

Here we describe our datasets in more detail.

**LSUN Church (18)** consists of 126,227 images of outdoor churches. The images are in the dataset are 256px on the short side. During training, 256×256 cropped images are used. A separate validation set of 300 images is used for comparisons against baselines.

**LSUN Bedroom (18)** consists of 3,033,042 images of indoor bedrooms. Like LSUN Church, the images are trained at 256×256 resolution. The results are shown with the validation set.

**Flickr Faces HQ (8)** consists of 70,000 high resolution aligned face images from `flickr.com`. Our model is initially trained at $512\times512$ resolution, and finetuned at 1024 resolution. The dataset designated 10,000 images for validation, but we train our model on the entire 70,000 images, following the practice of StyleGAN (8) and StyleGAN2 (9). For evaluation, we used randomly selected 200 images from the validation set, although the models are trained with these images.

**Animal Faces HQ (2)** contains a total of 15,000 images equally split between cats, dogs, and a wildlife category. Our method is trained at $256\times256$ resolution on the combined dataset without domain labels. The results are shown with a separate validation set.

**Portrait2FFHQ** consists of FFHQ (8) and a newly collected 19,863 portrait painting images from `wikiart.org`. The model is trained at $512\times512$ resolution on the combined dataset. The results of the paper are generated from separately collected sample paintings. We did not check if the same painting belongs in the training set. The test photographs are from CelebA (12). All images are aligned to match the facial landmarks of FFHQ dataset.

**Flickr Waterfall** is a newly collected dataset of 90,345 waterfall images. The images are downloaded from the user group "Waterfalls around the world" on `flickr.com`. The validation set is 399 images collected from the user group "*Waterfalls*". Our model is trained at $256\times256$ resolution.

**Flickr Mountains** is a newly collected dataset of 517,980 mountain images from Flickr. The images are downloaded from the user group "Mountains Anywhere" on `flickr.com`. For testing, separately downloaded sample images were used. Our model is trained at $512\times512$ resolution.

# 3 Detectability of Generated Images

To partially mitigate the concern of misusing our method as a tool for generating deceiving images, we see if a fake image detector can be applied to the images produced by our method. We run the off-the-shelf detector from (16), specifically, the Blur+JPEG(0.5) variant on the full, uncropped result images from this paper, and evaluate whether they are correctly classified as "synthesized". The results are shown in Table 3. For the most sensitive category, FFHQ faces, both previous generative models and our method have high detectability. We observe similar behavior, albeit with some drop-off on less sensitive categories of "church" and "waterfall".

| Method | Task | Dataset | | | |
|---|---|---|---|---|---|
| | | Church | FFHQ | Waterfall | Average |
| Im2StyleGAN (1; 8) | reconstruct | 99.3 | 100.0 | 92.4 | 97.2 |
| | swap | 100.0 | 100.0 | 97.7 | 99.2 |
| StyleGAN2 (9) | reconstruct | 99.7 | 100.0 | 94.4 | 98.0 |
| | swap | 99.8 | 100.0 | 96.6 | 98.8 |
| Swap Autoencoder (Ours) | reconstruct | 93.6 | 95.6 | 73.9 | 87.7 |
| | swap | 96.6 | 94.7 | 80.4 | 90.5 |

Table 3: **Detectability.** We run the CNN-generated image detector from Wang et al. (16) and report average precision (AP); chance is $50\%$. The CNN classifier is trained from ProGAN (7), the predecessor to StyleGAN (8). Because our method shares architectural components, a classifier trained to detect a different method can also generalize to ours, with some dropoff, especially for the waterfall class. Notably, the performance on FFHQ faces remains high. However, performance is not reliably at $100\%$ across all methods, indicating that future detection methods could potentially benefit from training on our method.