[Reviews · NeurIPS 2020]

Review 1

Summary and Contributions: This paper focuses on the task of image manipulation, and proposes a swapping autoencoder. Different from the recent mehtods, which directly synthesize the image from conditional input or look for meaningful editing operations in the existing latent space, Swap Autoencoder does it by disentangling one image into the structure and texture components and swapping them for mixing. Several potential applications have been shown to demonstrate its power.

Strengths: 1. From the results shown in the paper, it seems that the proposed method can disentangle the structure and texture quite well. 2. Several different applications have been tested and show plausible results.

Weaknesses: 1. The main idea of this paper is disentangling structure and texture by using an auto-encoder like structure. However, it is not a new idea and indeed studied in many previous methods. Though the authors try to differentiate this method to existing methods from the supervise/unsupervise aspect in the related work part, it is still not technically impressive to me. Moreover, there is no comparison to these disentangle methods. Maybe these methods cannot be directly applied to the tasks mentioned in this paper, but I think it should not be difficult. Even if the proposed method can produce better results, then what is the key ingredient? In this paper, the authors have not clearly explain what is the key and novel component that produce the good results. In fact, all the technical components look very common to me. 2. In this paper, the texture information is only encoded as a global vector, but for many natural images, they often contain very complex and diverse textures even within a single image. I do not think a global vector can encode such rich textures in a complete way. In fact, the authors only conduct their experiments on very simple datasets that only contain a semantic category, such as face, building or bedroom and the transferred textures look almost identical. Typical examples are Figure 4, the texture on one building is almost identical and the ground texture in the second bedroom case is not successfully transferred(first case transfer the sheet texture to the ground incorrectly). To summarize, I highly suspect this method cannot be applied to general natural images. If true,this is one big weakness when compared existing general texture transfer methods. 3. Many disentangle papers in neural style transfer and image translation are not cited and compared. "Disentangling content and style via unsupervised geometry distillation" "Style and Content Disentanglement in Generative Adversarial Networks" "StyleBank: An Explicit Representation for Neural Image Style Transfer" "Unsupervised Robust Disentangling of Latent Characteristics for Image Synthesis" 4. There are many artifacts in the results when zooming the local regions. For example, the grass texture is transferred to the building wall in the third example of Figure 1, the sheet textures are transferred to the ground in Figure 4. Considering the local artifacts and the default resolution except FFHQ is all set as 256 in the experiment part, I highly recommend the authors to provide high resolution results and demonstrate the capacity of the proposed method. 5. There are some spelling mistakes in the paper, like "latent spade" in line 185.

Correctness: Yes, the method is correct.

Clarity: The paper is well weitten.

Relation to Prior Work: Yes, it is clear.

Reproducibility: Yes

Additional Feedback: ************************** after rebuttal****************** I have carefully read the rebuttal, I am overall satisfactory with the authors' response. But as the authors said, current version has not positioned their main contribution or key component clearly. Besides, for the second limitation I mentioned in the original review, the authors have not explained it much, so I keep my first guess that a global latent code is not enough. Considering all the factors, I am more inclined to keeping my previous score. Of course, I am also okay if AC think the existing version of this paper is fine for acceptance.


Review 2

Summary and Contributions: This work tackles one of the problems of conditional image synthesis and controllable image manipulation. Namely, authors suggest a method that can blend “textures” of one image with a “structure” of another image under the condition that both images have structure of the same kind (both images are faces, or churches, or landscapes). This paper contributes to the field of generative image synthesis by significantly improving the visual quality of the results by leveraging modern network architectures(StyleGANV2) and smart problem formulation in terms of losses.

Strengths: 1) Superb visual quality of the results. 2) Clear writing 3) Thorough and extensive experiments. 4) New datasets: mountains and waterfalls. (minor factor, obtained from Flickr) 5) It has been shown that image can be effectively split into two representations and that structural representation can be as small as H/16 X W/16 X 8. Exactly the fact that we need only 8 channels to encode any structural detail is surprising to me.

Weaknesses: My main concern is lacking novelty of the method. The main declared technical novelty is the co-occurence GAN loss introduced in in Eq. 4 is not as novel as it seems, I elaborate my concern below. In general this paper resembles a lot papers like MUNIT[26] or FUNIT[47] which also have tackled the problem of splitting image into structure and texture (called “content” and “style” in FUNIT and MUNIT papers) and blending them back together. But in this work superb image quality is achieved thanks to using architecture and tricks introduced in StyleGAN2[38] paper. Authors write in ll.55-57 that opposite to the FUNIT[47] their method can be applied on the fly but actually FUNIT[47] is also designed to work with unknown style examples. Moreover, if one look into the main novelty of the paper (co-occurence GAN loss in Eq.4) it is pretty similar to the approach suggested in the FUNIT[47] paper. Namely, to train the patch discriminator authors first cut out multiple reference patches from a “texture” image and one query patch: either from real image or synthesized result. Each patch is individually processed by the PatchEncoder (Fig2. in Supp). Afterwards, those latent codes of reference patches obtained by PatchEncoder are averaged and concatenated with patchEncoder output of the query patch. This operation is very similar to the style approximation(by averaging latent codes) for unseen class in FUNIT[47] paper in Eq. 6. The main difference is that FUNIT[47] uses different style images as reference patches and this work crops out reference patches from the same “texture” image. Thus, it would be nice to see how well does the model perform on other datasets ubiquitous in the field of conditional image synthesis like: Foods. Paper: Automatic expansion of a food image dataset leveraging existing categories with domain adaptation. Flowers. Paper: Automated flower classification over a large number of classes Birds. Paper: Building a bird recognition app and large scale dataset with citizen scientists: The fine print in fine-grained dataset collection.

Correctness: Though results are pleasing to the eye, there are still some issues with the methodology and evaluation: 1) Tab.1 compares suggested method to the StyleGAN2. However, suggested method starts from real images induced with real structure and real texture. StyleGAN2 has to generate it from a random noise vector. Additionally StyleGAN2 was not optimized to take latent codes from different images, blend them and yield realistic results. To me this evaluation is not fully fair. 2) One limiting factor of the method can be its inability to alter structure if texture requires it. I mean the following scenario that appears in the field of conditional image synthesis: if we take a daisy flower as a texture and a lily flower as a structure - we may end up with a fluffy lily of weird color. This limitation of the method should be stressed in the paper.

Clarity: The writing is clear and concise and not overwhelmed with technical details. Yet, non-trivial tricks and techniques used for training are mentioned in the text.

Relation to Prior Work: In my opinion, this work should focus more on papers like FUNIT[47], StarGAN[11] and similar. And, as discussed in the Weaknesses section, though this work looks novel, under the hood it is heavily inspired by the FUNIT[47] paper.

Reproducibility: Yes

Additional Feedback: How long does it take to train a model? Can one use pretrained StyleGAN2 as a decoder?


Review 3

Summary and Contributions: The paper proposes a method for disentangling the structure and content of images which allows for various forms of image manipulation. An encoder network converts an input image into structure and texture codes; the former is a convolutional feature map and the latter is a vector. A decoder network inputs the structure and texture codes and generates an RGB image. The networks are trained jointly to (1) reconstruct input images; (2) generate realistic images (as judged by a discriminator) given either the structure and texture codes from the same image or when swapping the codes from a pair of images; (3) generate images with similar patch-level statistics when given the same structure codes (as judged by a patch discriminator). After training, swapping the texture and structure codes of images allows for a sort of structure-preserving style transfer; interpolating between texture codes allows for continuous style transfer, and masked computation allows users to edit specified regions of images. Experiments are performed on several datasets, and demonstrate that the method can both reconstruct images and can generate realistic composite images (using large-scale user studies).

Strengths: This is a strong paper overall. The method builds on recent advances in generative models to improve the degree to which end users can control the generated images. The method makes sense, the qualitative results are impressive, and the experiments are thorough and convincing. Results are validated on several datasets, using both automated metrics and several user studies. In addition to testing on several standard datasets used for generative models, the authors also introduce new datasets of portraits and waterfalls; I hope that the authors can release these datasets. Overall the paper is also very well-written and easy to follow.

Weaknesses: COMPARISON WITH UNIVERSAL STYLE TRANSFER On the whole the problem statement is very similar to recent work on arbitrary / universal style transfer, which inputs a “content” and “style” image to a single feedforward network which generates an image combining both; for example [A1-A7]. The paper largely ignores this active area of recent work; it should be discussed more explicitly, and the paper’s contributions should be put into context with respect to this area of work. The WCT^2 method is one example of photorealistic universal style transfer; from Table 1 WCT^2 is similar to the proposed method in terms of perceptual fooling rates, and from Figure 6 WCT^2 actually outperforms the proposed method in terms of perceptual structure preservation. Thus one of the main improvements of the proposed method over WCT^2 is runtime (Figure 3, and Table 1). However WCT^2 is not the best representative of universal style transfer methods, both because it emphasizes photorealism and because it is much slower than fully feed-forward approaches; for example timing information in [A6] shows that it is roughly the same speed as the method proposed by this paper. While I appreciate the comparisons with WCT^2, I think that the method should also have been compared with at least one recent real-time approach to non-photorealistic universal style transfer such as [A6] or [A7]. [A1] Ghiasi et al, “Exploring the structure of a real-time, arbitrary neural artistic stylization network.”, BMVC 2017 [A2] Huang and Belongie, “Arbitrary Style Transfer in Real-time with Adaptive Instance Normalization”, ICCV 2017 [A3] Li et al, “Universal Style Transfer via Feature Transforms”, NeurIPS 2017 [A4] Sanakoyeu et al, “A Style-Aware Content Loss for Real-time HD Style Transfer”, ECCV 2018 [A5] Mechrez et al, “The Contextual Loss for Image Transformation with Non-Aligned Data”, ECCV 2018 [A6] Li et al, “Learning Linear Transformations for Fast Image and Video Style Transfer”, CVPR 2019 [A7] Kotovenko et al, “Content and Style Disentanglement for Artistic Style Transfer”, ICCV 2019 INJECTIVITY I don’t think that the claims around injectivity are correct (L118 - L126). Specifically, though the reconstruction loss encourages injectivity over the data distribution, the architecture of E means that it cannot represent an injective function. From L11-12 of the Supplementary, the encoder maps an image of shape (256, 256, 3) to a structure code of shape (16, 16, 8) and a texture code of shape (1, 1, 2048), so considered as a function from R^N to R^M it converts a list of N = 256 * 256 * 3 = 196,608 numbers into a list of M = 16 * 16 * 8 + 2048 = 4096 numbers; moreover E is a continuous function, since it is a composition of continuous primitives (convolution, LeakyReLU, etc). By the following Lemma, E cannot also be injective: Lemma A: If N > M, there are no continuous injections from R^N to R^M. Proof: Suppose that f: R^N -> R^M is a continuous injection. Since N > M, the M-dimensional sphere S^M is a subset of R^N; let g: S^M -> R^M be the restriction of f to S^M. Then g is a continuous injection from S^M to R^M, so by the Borsuk-Ulam Theorem there exists x \in S^M such that g(x) = g(-x); thus g is not injective, and so f cannot be injective. This is a contradiction. Interestingly, Terry Tao has a blog post presenting an alternate proof of this result in terms of Brouwer’s fixed point theorem: https://terrytao.wordpress.com/2011/06/13/brouwers-fixed-point-and-invariance-of-domain-theorems-and-hilberts-fifth-problem/ I don’t believe that this contradicts Lemma 1 from the Supplementary: I believe but am not completely sure that the preconditions of Lemma 1 can only hold if X and Z are subsets *of the same dimension*.

Correctness: Yes, except for the claims around injectivity above.

Clarity: The paper is very well-written and easy to follow. The main text is self-contained, and the supplementary material provides many additional details and results.

Relation to Prior Work: The paper omits critical references and comparisons to recent work on universal and arbitrary style transfer; see comments above.

Reproducibility: Yes

Additional Feedback: On the whole this is a strong paper. There is an incorrect claim about the injectivity of the encoder, but this can be removed without affecting the overall results. More importantly, the main weakness of the paper is that it omits critical references and comparisons to recent work on universal style transfer. In the rebuttal I would like the authors to comment on connections and comparisons between these methods and their work. I am also concerned about the fact that the Broader Impacts section of this paper contains an additional figure showing additional experimental results and an additional application of the method; this seems to me like a violation of the paper formatting guidelines, which state: “Submissions are limited to eight content pages, including all figures and tables; additional pages containing a statement of broader impact, acknowledgements and funding disclosures, and references are allowed.” On the whole I lean slightly towards acceptance, since the method is novel and interesting, the results are convincing, and the paper is well-written. I would feel more positively about the paper if the authors could provide a more convincing discussion or comparison of their results with recent work on universal style transfer. Note also that my rating does not reflect the potential violation of the paper formatting guidelines; I defer to the AC for this issue. =================UPDATE=================== I have read the other reviews and the author response; the author response addresses all of my concerns with the paper (except the possible violation of paper formatting guidelines, to which I defer to the AC). I agree with R1 that this paper is not the first to disentangle style and content, and agree that (especially the first two) additional references provided by R1 ought to have been cited. However I agree with the authors that their work is similar but not identical to these prior methods, and furthermore that the submission has significantly more impressive qualitative results than these prior approaches. I can see the similarity with FUNIT that R3 is concerned about, but I also agree with the authors that their work is again clearly novel and distinct from this prior work, so I disagree with R3 that the similarity with FUNIT is a black mark against this paper. Overall I still think that this is a strong paper with clear writing and impressive results, and a clear accept.

[Author Response · NeurIPS 2020]

We sincerely thank our reviewers for the constructive feedback. We are glad to see our paper is well-written (R1, R3, R4), has strength in visual quality (R1, R3, R4), and performs diverse (R1) and thorough (R3, R4) experiments.

**Novelty [R1, R3]** Note that we do not claim to be the first to use swapping for disentanglement. Our key contribution is in the way we *constrain* this disentanglement, by separating intra-image information (co-occurrence of patches within the same image) from inter-image information (visual content across images within the dataset). Typically, learning for image synthesis happens by gathering information across the entire image dataset (or across an image class). However, this is difficult to do correctly, as the texture statistics within each image can be quite different and incompatible. Thus the recent popular "internal learning" methods, such as InGAN[58] or SinGAN[59], show superior results by learning from the patches of just the input image, utilizing the commonality of texture statistics. But, of course, methods like [59] can't learn that much because their dataset is very limited (just one image!). What our proposed method is doing is disentangling the texture-like information useful for internal learning from the structure-like information that can profit from external learning. To separate out internal learning from external learning, we model internal information (texture code) using patch co-occurrence D (which operationalizes Julesz's 2nd order texture conjecture), which leaves the structure code representing texture-invariant (compatible with texture code of any image). So, we combine the best of both internal and external learning. The swapping is just a self-supervised pretext task to accomplish this disentanglement. Figure 6 confirms that the quality of disentanglement is superior to generative models that don't utilize internal information (Im2StyleGAN[1], StyleGAN2[38]) or external information (STROTSS[41], WCT$^2$[69]). In the current draft, this motivational story got distributed throughout the paper, but in the next draft we will be sure to summarize it concisely in the intro as well.

**Comparison to FUNIT [R3]** FUNIT is fundamentally different from our work; it requires labels to define the disentanglement of style from structure. For example, the AnimalFaces, Birds, Flowers, and Food datasets contain 149, 444, 85, and 224 classes, respectively. ***For this reason, FUNIT cannot be trained on any of the datasets in our paper***. In contrast, our method is fully unsupervised, and works because co-occurrence statistics of patches of a single image carry enough signal for learning a smooth embedding space for image editing. By doing so, our method can be easily trained on user-collected datasets such as the Flickr Mountains and Waterfalls, which is useful for long-tail image editing. In more detail, the learning objective and discriminator design are substantially different; the FUNIT discriminator requires class labels, while ours performs internal learning on unlabeled images (see Novelty). While substantially different to FUNIT's discriminator in concept, we agree our discriminator shares architectural similarities with FUNIT's encoder. To clarify, the PatchEncoder of Figure 2 of SuppMat is the feature extractor of our discriminator, not our encoder. We apologize for any confusion.

**Image Quality, Resolution, and Versatility [R1]** We show our method works well on a wider variety of datasets than most GAN-based methods, on human and animal faces, outdoor and indoor scenes, user-collected datasets, and paintings (Section 1.3 of Supp Mat). Moreover, our method supports HD resolution (e.g. the mountain example of Figure 7, the demo video and Supp Mat). Lastly, we evaluated the quality of texture transfer against two SOTA style transfer methods (non-photorealistic and photorealistic), and show that our method is competitive, if not better. Therefore, we believe our method has advantage over general texture transfer methods.

**Cite and compare to disentanglement / neural style transfer papers [R1]** Thank you for the suggestions. We will cite the suggested works. Our end goal differs in that a real, existing image must be faithfully embedded for editing, while the suggested works do not focus on accurately reconstructing input images.

**Comparison to StyleGAN2 [R3]** Our method aims to build an image editing model in an unsupervised setting. To our knowledge, the state-of-the-art unsupervised methods for GANs-based image editing are Im2StyleGAN[1] and StyleGAN2[38]. Randomly-generated StyleGAN2 images cannot be used for this application.

**Injection [R4]** We appreciate your feedback. We will modify "clearly injective (L119)" to "optimized toward injection", and also revise the Supp Map accordingly.

**About Broader Impact [R4]** We are happy to move the figure to Supp Mat based on reviewers/AC's recommendation.

**Structural Limitation [R3]** We show the result of the suggested lily / daisy example. Please see the figure below.

input structure | input style | output (ours) | Li et al.,19 (artistic) | Li et al.,19 (photorealistic)

Structure-style mixing results on the Oxford Flower dataset. Our model can modify fine-level structure if the texture code requires it, while maintaining the coarse-level structure. We additionally show the result of a real-time style transfer method (Li et al., CVPR2019 [A6 suggested by R4], both artistic and photorealistic version) that makes only small changes.

**Comparison to recent style transfer models [R4]** Thank you for the suggestion; we will discuss our work in the context of neural style transfer more explicitly. Note that style transfer methods, even if real-time, are not suitable for *natural* image manipulation, because they often fail to produce photorealistic images (please see left) and do not allow controllable structural editing. For thoroughness, in the submission, we compared the quality of disentanglement to both SOTA non-photorealistic (STROTSS) and photorealistic (WCT$^2$) style transfer methods. While WCT$^2$ does achieve better structural preservation, it does so by barely changing the style (Figure 5).

[Meta-Review · NeurIPS 2020]

*The paper uses the broader statement section to report additional results. This is not acceptable and some papers have been desk-rejected for doing the same thing. These additional results have to be moved elsewhere.* The reviewers' opinions on this paper diverge even after considering the rebuttal and discussing. The meta-review is thus unusually detailed. The paper proposes an approach to image editing by disentangling the structure and texture using an autoencoder with the latent space decomposed into two parts - corresponding to texture and structure. When both are taken from the same image, the model is required to reconstruct the image. However, when the texture representation is taken from a different image, the model is required to generate a realistic image with local statistics similar to the texture image (enforced via a patch-based discriminator). The method demonstrates good qualitative and quantitative results on several datasets. Pros: 1) High quality qualitative results (R3, R4). Many of these are presented in the supplementary material. For a paper on image editing this is crucial. 2) Good quantitative results, largely based on user studies (R4) 3) Thorough and convincing experiments (R3, R4) 4) Clear presentation (R3, R4) 5) Fast runtime compared to some of the competing methods Cons: 1) Limited novelty relative to prior methods, e.g. FUNIT (R1, R3) 2) Encoding texture as a fixed-length vector may not be sufficient to represent complex textures (R1) the choice of the representation of the texture information (R1) 3) Limited comparison to (photorealistic) style transfer methods (R4) 4) Missing references (R1) Limited novelty is the biggest concern, but I believe even though the components are quite known, this combination of them is new, intuitive, clean, and works well. Moreover, two of the reviewers have been quite satisfied with the authors' rebuttal and the third, most negative, reviewer (R3), has not responded to the rebuttal and has not participated in the discussion. After carefully considering the reviews, the rebuttal, and the paper itself, I recommend acceptance. Authors are, however, required to remove additional results from the broader statement section - that's not what the section is intended for. Moreover, the authors are encouraged to address the reviewers' concerns in the final version of the paper.